# Close Cardiovascular Monitoring during the Early Stages of Treatment for Patients Receiving Immune Checkpoint Inhibitors

**DOI:** 10.3390/ph17070965

**Published:** 2024-07-21

**Authors:** Danielle Delombaerde, Christof Vulsteke, Nico Van de Veire, Delphine Vervloet, Veronique Moerman, Lynn Van Calster, Anne-Marie Willems, Lieselot Croes, Félix Gremonprez, Astrid De Meulenaere, Ximena Elzo Kraemer, Kristien Wouters, Marc Peeters, Hans Prenen, Johan De Sutter

**Affiliations:** 1Integrated Cancer Center Ghent, Department of Medical Oncology, AZ Maria Middelares, 9000 Ghent, Belgium; christof.vulsteke@mijnziekenhuis.be (C.V.); felix.gremonprez@mijnziekenhuis.be (F.G.); astrid.demeulenaerde@mijnziekenhuis.be (A.D.M.); ximena.elzokraemer@mijnziekenhuis.be (X.E.K.); 2Center for Oncological Research (CORE), Integrated Personalized and Precision Oncology Network (IPPON), University of Antwerp, 2610 Wilrijk, Belgium; lieselot.croes@uza.be (L.C.); marc.peeters@uza.be (M.P.); hans.prenen@uza.be (H.P.); 3Department of Cardiology, AZ Maria Middelares, 9000 Ghent, Belgium; nico.vandeveire@mijnziekenhuis.be (N.V.d.V.); delphine.vervloet@mijnziekenhuis.be (D.V.); veronique.moerman@mijnziekenhuis.be (V.M.); lynn.vancalster@mijnziekenhuis.be (L.V.C.); anne-marie.willems@mijnziekenhuis.be (A.-M.W.); johan.desutter@mijnziekenhuis.be (J.D.S.); 4Department of Cardiology, Free University Brussels, 1000 Brussels, Belgium; 5Multidisciplinary Oncologic Center Antwerp (MOCA), Antwerp University Hospital, 2650 Edegem, Belgium; 6Antwerp University Hospital, Clinical Trial Center (CTC), CRC Antwerp, 2650 Edegem, Belgium; kristien.wouters@uza.be; 7Faculty of Medicine and Health Sciences, Ghent University, 9000 Ghent, Belgium

**Keywords:** immune checkpoint inhibitor, immune-related adverse event, cardiac troponin, myocarditis, subclinical cardiotoxicity, diastolic function

## Abstract

**Background:** There is an unmet medical need for the early detection of immune checkpoint inhibitor (ICI)-induced cardiovascular (CV) adverse events due to a lack of adequate biomarkers. This study aimed to provide insights on the incidence of troponin elevations and echocardiographic dynamics during ICI treatment in cancer patients and their role as potential biomarkers for submyocardial damage. In addition, it is the first study to compare hs-TnT and hs-TnI in ICI-treated patients and to evaluate their interchangeability in the context of screening. **Results:** Among 59 patients, the mean patient age was 68 years, and 76% were men. Overall, 25% of patients received combination therapy. Although 10.6% [95% CI: 5.0–22.5] of the patients developed troponin elevations, none experienced a CV event. No significant changes were found in 3D left ventricular (LV) ejection fraction nor in global longitudinal strain f (56 ± 6% vs. 56 ± 6%, *p* = 0.903 and −17.8% [−18.5; −14.2] vs. −17.0% [−18.8; −15.1], *p* = 0.663) at 3 months. There were also no significant changes in diastolic function and right ventricular function. In addition, there was poor agreement between hs-TnT and hs-TnI. **Methods:** Here, we present a preliminary analysis of the first 59 patients included in our ongoing prospective clinical trial (NCT05699915) during the first three months of treatment. All patients underwent electrocardiography and echocardiography along with blood sampling at standardized time intervals. This study aimed to investigate the incidence of elevated hs-TnT levels within the first three months of ICI treatment. Elevations were defined as hs-TnT above the upper limit of normal (ULN) if the baseline value was normal, or 1.5 ≥ times baseline if the baseline value was above the ULN. **Conclusions:** Hs-TnT elevations occurred in 10.6% of the patients. However, no significant changes were found on 3D echocardiography, nor did any of the patients develop a CV event. There were also no changes found in NT-proBNP. The study is still ongoing, but these preliminary findings do not show a promising role for cardiac troponins nor for echocardiographic dynamics in the prediction of CV events during the early stages of ICI treatment.

## 1. Introduction

Over the past decade, immune checkpoint inhibitors (ICIs) have brought significant advantages to the field of oncology as they have demonstrated a survival benefit in various cancer types [1,2]. As survival increases, patients will become more susceptible to cardiac adverse events [3]. ICIs, like other cancer therapies (e.g., chemotherapy, tyrosine kinase inhibitors, vascular endothelial growth factor inhibitors, radiotherapy), can lead to cancer treatment-related adverse events. Cardiovascular (CV) immune-related adverse events (irAEs), in particular, have become a topic of interest, due to the high rates of morbidity and mortality associated with ICI-induced myocarditis [4]. Although myocarditis was the first reported cardiac irAE, many other cardiac irAEs have also been reported [5,6]. Nevertheless, CV irAEs are still poorly understood and have most likely been under-reported during the last few years, due to the lack of routine cardiac screening (echocardiography, electrocardiography, and troponins), wide varieties in clinical presentation, and the scarcity of real-world studies in patients receiving ICI therapy [5,7,8,9,10,11,12].

Certain guidelines (ASCO and SITC) do not recommend systematic cardiac biomarker testing, whereas others (ESC and NCCN) suggest its consideration [13,14,15,16]. However, these guidelines are based on limited evidence and expert opinions, due to a paucity of real-world data [16,17]. Furthermore, the classification system used for grading adverse events in cancer patients, i.e., the Common Terminology Criteria for Adverse Events (CTCAE, Version 5), suffers from different limitations regarding cardiac adverse events [18]. First of all, there is no distinction between high-sensitivity troponin T (hs-TnT) and I (hs-TnI) [19]. Second, the threshold for a ‘positive’ troponin level remains unclear. Third, N-terminal brain natriuretic peptide (NT-proBNP) is also considered to be a potential biomarker for predicting cardiac adverse events [20,21]. However, it is not listed in the current CTCAE. Thus, the present classification systems are insufficient for assessing deviations in cardiac biomarkers, which are often determined in the detection of cancer treatment-related adverse events.

Following our narrative review, we initiated a prospective, multicenter trial, i.e., ‘Extensive cardiovascular characterization and follow-up of patients receiving ICIs’ (NCT05699915) [22,23,24,25,26]. In this trial, we systematically assess different cardiac biomarkers as well as 3D echocardiography (baseline, 3, 6, 12, and 24 months) in cancer patients with a solid tumor treated with ICIs. Since our study was initiated, other research groups also started publishing their results [8,9,10,11,12,27,28,29,30,31,32,33,34]. However, these studies differ, as they often measure either hs-TnI (majority) or hs-TnT, only take baseline measurements into account, focus primarily on myocarditis, do not always include systematic echocardiography, lack statistical analysis, and include patients who receive ICIs in combination with other potentially cardiotoxic systemic anticancer regimens.

Here, we present the results of a preliminary analysis of the first cohort of 59 patients after three months follow-up. The main purpose of this preliminary analysis is to provide additional insights on the incidence of troponin elevations upon routine monitoring in ICI-treated patients and to explore TTE values that could possibly identify submyocardial damage in a uniform, Caucasian cohort. In addition, it is the first study to compare hs-TnT and hs-TnI in ICI-treated patients and evaluate their interchangeability in the context of screening.

## 2. Results

### 2.1. Study Population

The mean patient age was 68 ± 12 years, and 76% were male. The most commonly used ICI was pembrolizumab (37%), followed by combination therapy, i.e., nivolumab–ipilimumab (25%). Bladder cancer, melanoma, and renal cell cancer were among the most frequent cancer types. Arterial hypertension, hypercholesterolemia, and diabetes mellitus type 2 were present in 46%, 66%, and 17% of the patients, respectively. In total, ten out of 54 patients had coronary artery disease at baseline, while 63% were either former or current smokers (Table 1).

### 2.2. Cardiac Biomarkers: Hs-TnT, Hs-TnI, and NT-ProBNP

Troponin T levels were measured prior to each ICI cycle during the first three months of treatment. The cumulative incidence of hs-TnT elevations was 10.6% [95% CI: 5.0–22.5] (Figure 1). Thirty-six patients had normal (<14 ng/L) hs-TnT levels, while 23 patients had elevated (≥14 ng/L) levels at baseline. Of the patients with elevated baseline levels 9, 19, and 3 had coronary artery disease, chronic kidney disease (ranging from grade 2 to 3B) and heart failure, respectively. In total, 5 out of 36 patients developed elevated troponin levels within the first three months. One patient, who had elevated levels at baseline, also met the primary endpoint. Three patients did not have a history of CV disease, whereas two patients did. The other patient had a history of chronic obstructive pulmonary disease. Death was accounted for as a competing risk factor, as six patients died within the first 3 months of treatment, of which four had no elevations and two did. Three patients were followed-up for less than 90 days. Despite hs-TnT elevations, none of the patients experienced a CV event.

Troponin I and NT-proBNP levels, on the other hand, were only measured at baseline and three months. As opposed to Hs-TnT, none of the patients with normal baseline hs-TnI developed hs-TnI elevations (Table 2). One patient had levels above the ULN at three months; however, this was already present at baseline. A total of 21 out of 49 patients had NT-proBNP levels higher than the ULN at baseline and three months. Two patients, who had normal NT-proBNP levels at baseline, developed elevations during the first three months of treatment. The 3-month blood sample was only available for 49 out of 59 patients (Table 2) (Appendix A).

### 2.3. Agreement between hs-TnI and hs-TnT Elevations

Cohen’s κ was run to determine if there was an agreement between elevations in hs-TnT and hs-TnI. There was only poor agreement between elevations in both troponins at baseline, κ = 0.155, *n* = 59, [95% CI: 0.006–0.316], and *p* = 0.026, and at three months, κ = 0.082, *n* = 49, [95% CI: 0.071–0.235], and *p* = 0.147. All patients with elevated hs-TnI at baseline had elevated hs-TnT levels. Contrarily, only 3/23 patients with elevated TnT also had elevated TnI levels. NT-proBNP levels did not significantly differ (131 pg/mL [53; 330] vs. 119 pg/mL [53; 696], *n* = 49, *p* = 0.521) (Table 3).

### 2.4. Echocardiography Parameters at Baseline and Three Months 

Only 50 out of 59 patients received their 3-month TTE. Two were treated in the best supportive care setting, for which their 3-month cardiology visit was canceled. One patient refused further CV follow-up shortly after treatment initiation, while six other patients died prior to their 3-month visit due to progressive disease (Appendix A). Furthermore, due to limited image quality and/or prior valve replacement, it was not possible to measure each TTE variable for all 50 patients (Table 4).

There was no change in 3D left ventricular ejection fraction (LVEF) after three months of ICI treatment (56 ± 6% vs. 56 ± 6%, *n* = 44, *p* = 0.903). Similar results were found for LV GLS (−17.8% [−18.5; −14.2] vs. −17.0% [−18.8; −15.1], *n* = 37, *p* = 0.663). RV function was assessed using TAPSE and s-wave; however, no significant differences were found (TAPSE 23 ± 5 mm vs. 22 ± 4 mm, *n* = 48, *p* = 0.335; s-wave 12 ± 3 cm/s vs. 13 ± 3 cm/s, *n* = 47, *p* = 0.578). After ICI initiation, the LA area did not dilate (17 ± 4 cm^2^ vs. 18 ± 5 cm^2^, *n* = 47, *p* = 0.264). There were also no significant changes in the subgroup with coronary artery disease at baseline (Appendix A). None of the patients experienced a CV event during the first three months of treatment.

## 3. Discussion

In the present study, we assessed cardiac biomarkers along with routine 3D echocardiography in ICI-treated patients. In our cohort of 59 patients, we found that: (1) 10.6% developed hs-TnT elevations, in the absence of CV events; (2) almost half of the patients had elevated hs-TnT and NT-proBNP levels at baseline; (3) hs-TnT and hs-TnI showed poor agreement; (4) no significant changes were found on 3D echocardiography nor in NT-proBNP levels at three months.

Cardiac biomarkers play a key role in the diagnosis of CV disease in non-cancer patients. While troponins I and T are biomarkers of myocardial injury [35,36], (NT-pro)BNP marks increased wall stress upon elevation [37]. Previous research has demonstrated the beneficial role of measuring these markers in other cardiotoxic anti-cancer therapies, such as anthracyclines [38]. Petricciuolo et al. [24] and Waissengein et al. [32], on the other hand, showed that baseline troponin levels can predict future MACEs. However, as some guidelines have also recommended serial monitoring, we aimed to evaluate the role of these biomarkers during treatment. In our cohort of 59 patients, approximately half already had troponin T or NT-proBNP levels above the ULN at baseline, while elevated hs-TnI levels were only present in one patient. Similar results were reported by Kurzhals et al. [30]. Asymptomatic troponin elevations in cancer patients have previously been linked to disease progression, other (cardiac) comorbidities, and/or the deterioration of the patient’s clinical status [39]. In addition, most patients received prior oncological treatment, which could also have contributed to elevated baseline levels. During treatment, 10.6% developed hs-TnT elevations. Notably, there were no clinical CV events in any of the patients. This finding is in line with the results found in a sub-analysis of the JAVELIN 101 trial, a phase 3 trial of advanced renal cell cancer patients treated with a combination of a tyrosine kinase inhibitor and an anti-PD-L1 antibody, in which the routine monitoring of cardiac biomarkers in asymptomatic patients was not useful for the early detection of CV irAEs [9]. Unlike the patients in the JAVELIN 101 trial, we assembled a uniform cohort of patients who received ICIs in the absence of other systemic anti-cancer regimens.

While hs-TnT and hs-TnI elevations have a good biochemical concordance in patients with acute coronary syndromes, their role in the prediction and screening of CV irAEs remains unclear [40]. Hs-TnI is often preferred above hs-TnT as it has been perceived to be more cardio-specific than hs-TnT. The reason for this discrepancy between troponin I and T still remains unclear. As previously mentioned, the majority of studies measure either hs-TnT or hs-TnI, resulting in limited data on measurements of both troponins within the same cohort. This is the first study to prospectively evaluate the agreement between hs-TnT and hs-TnI elevations in cancer patients receiving ICI therapy. We only found a poor agreement between both troponins at baseline and at three months. Our results are similar to the ones reported in a general population cohort [19]. However, we did not perform a sub-analysis based on CV risk factors. Furthermore, since none of the patients developed a CV event, we were unable to compare these levels in the context of cardiotoxicity. However, a recent study did show that in patients who were hospitalized for symptomatic ICI myocarditis (*n* = 60), hs-TnI levels normalized earlier on than hs-TnT, suggesting that hs-TnT could be of superior clinical utility [41]. Nevertheless, further data are required to fully understand the role of hs-TnT and hs-TnI in the context of screening for CV events in ICI-treated patients.

Echocardiography is currently the preferred imaging technique for the diagnosis and management of myocardial damage and is recommended in moderate- and high-risk patients prior to ICI-treatment initiation. A TTE prior to ICI treatment in each patient, on the other hand, may be considered (level of evidence C in the European Society of Cardiology guidelines). In addition, routine TTEs during ICI treatment are currently not listed. In our study, all patients received a baseline and a 3-month TTE, including 3D LVEF, GLS (class I recommendation, level of evidence C),and an evaluation of LV diastolic function and RV systolic function [16]. GLS has previously demonstrated its efficacy in cardiology for identifying subtle left ventricular myocardial dysfunction in CV diseases [42,43]. As a result, research shifted towards GLS, since new strategies were needed for the early detection of cancer treatment-related cardiac adverse events to improve prognosis and patient outcomes; LVEF often lacks the sensitivity to detect early LV systolic impairment. Extensive research on the prediction and detection of CV events upon traditional cytotoxic chemotherapies illustrated that a decrease in GLS can serve as an early predictor of CV events and often precedes declines in LVEF [44,45,46]. The exact role of GLS in the routine follow-up of ICI-treated patients still remains a topic of controversy. Our results reflect those of Awadalla et al. [47] who also found no significant differences in GLS in their ICI-treated control group (*n* = 92, both pre- and on-ICI were only available for 14 patients) who did not develop myocarditis. However, it remains unclear at which specific timepoint on-treatment GLS was evaluated. Pohl et al. [48] also found no significant changes in LV GLS, LVEF, LV volumes, diastolic function, and TAPSE (*n* = 30) in patients with melanoma after one month of treatment (nivolumab or nivolumab/ipilimumab).

Contrarily, Mincu et al. [49] did find a significant reduction in GLS after only one month of treatment in a subgroup of 22 melanoma patients who developed non-cardiac irAEs. The discrepancy with our cohort could be attributed either to the fact that Mincu et al. [49] excluded patients with CV disease, which in turn complicates the future representativeness of GLS for a real-world ICI-treated population, or due to the fact that we did not take irAEs, other than cardiac, into account yet. Nishikawa et al. [33] also found a decrease in GLS in five out of the ten patients who developed myocardial injury, of which two had concomitant irAEs. Nevertheless, no statistical analyses were performed. In addition, Tamura et al. [11] found significant changes in deformation imaging in patients who developed troponin elevations (18/129). So far, the small sample size of our study has precluded subgroup analyses based on troponin elevations. Moreover, these findings are from a single center in Japan and cannot be extrapolated to our Caucasian patient cohort, as patient characteristics and tumor types differ [11]. Xu et al. [34] also reported the significant deterioration of LV GLS, as well as the RV function (RV GLS and TAPSE) within 220 days of treatment. Hence, RV dysfunction might develop earlier on than LV dysfunction. Notably, more than half of the patients did receive ICIs in combination with other systemic cancer treatments which could have also promoted myocardial injury. Furthermore, these values were investigated over an extended period of time. In our study, a longer follow-up is needed to confirm or challenge these results.

Our study has several limitations. It is a preliminary analysis of the first 59 patients included in our ongoing prospective trial. The sample size of the complete trial, i.e., a minimum of 276 patients, was not adjusted for this interim analysis, as study-level conclusions will only be made upon completion.

## 4. Methods

### 4.1. Study Population

All patients 18 years or older with a solid tumor eligible for and started with anti-PD-1, anti-PD-L1 and/or anti CTLA-4 treatment in mono- or combination therapy, and who signed informed consent, were included. Patients were excluded if they had received prior treatment with immunotherapy (ICIs, T-cell transfer therapy, cancer treatment vaccines or immune modulators). Patients receiving ICIs in combination with other systemic anti-cancer agents (chemotherapy, tyrosine kinase inhibitors, etc.) were excluded. The full eligibility criteria of the trial protocol are available online [50]. Patients were recruited from four different hospitals: Antwerp University Hospital, AZ Maria Middelares, AZ Sint-Elisabeth Zottegem, and AZ Sint-Vincentius Deinze.

The study was approved by the central Ethics Committee of the Antwerp University Hospital (2021–1908, 2022–1908) and follows the standards of the Declaration of Helsinki, in compliance with all national and local regulatory laws, and is consistent with the Good Clinical Practices guidelines. The protocol was also approved by the local Ethic Committees of AZ Maria Middelares, AZ Sint-Elisabeth Zottegem, and AZ Sint-Vincentius Deinze.

### 4.2. Medical History and Biochemical Parameters

Upon enrollment, the following data were collected from electronic medical records: informed consent, demographics, medical history, CV risk factors, oncological disease and stage, prior cancer history, prior/concomitant medication, cardiac biomarkers, and other relevant parameters. Troponins (hs-TnT or hs-TnI according to the site’s local practice) were measured for all participants at baseline and prior to each ICI cycle. An additional blood sample (serum) was taken at 3 months and temporarily stored in the biobank for the future determination of hs-TnT, hs-TnI and NT-proBNP [50].

Hs-TnT was measured on a COBAS apparatus (Roche Diagnostics; Mannheim, Germany, limit of detection 3 ng/L, ULN 14 ng/L). Hs-TnI was measured on an Atellica^®^ IM (Siemens Healthineers, New York, NY, USA; limit of detection 1.6 ng/L, ULN 45.2 ng/L). NT-proBNP levels were determined using Atellica^®^ IM (Siemens Healthineers, New York, USA; limit of detection 20 pg/mL, ULN 125 pg/mL < 75 years and 450 pg/mL for ≥75 years) and Elecsys cobas e 801 (Roche Diagnostics, Mannheim, Germany; limit of detection 5 pg/mL, ULN 125 pg/mL) (Appendix A).

### 4.3. Three-Dimensional Transthoracic Echocardiography

Three dimensional transthoracic echocardiography (TTE) was performed at baseline and at three months using a Vivid E95 ultrasound system (GE Healthcare, Horten, Norway) by a dedicated cardiologist. Systolic function, diastolic function, and ventricular and atrial geometry were assessed according to the American Society of Echocardiography and the European Association of Cardiovascular Imaging guidelines [51]. Full 3D data sets were acquired to evaluate left ventricular volumes and calculate 3D ejection fraction. Two-dimensional speckle tracking was used to perform the semi-automated deformation imaging of the left ventricular (LV) global longitudinal strain (GLS) using three apical views (4-, 2-, and 3-chamber). The tricuspid annular plane systolic excursion (TAPSE) and right ventricular (RV) free wall basic segment peak systolic velocity (s’-wave) using color coded tissue Doppler imaging were measured. The maximal left atrial area (LA) was measured on an apical 4-chamber view. All echocardiographic images were digitally stored on EchoPac workstation (GE Healthcare, Horten, Norway).

### 4.4. Study Endpoints

The primary endpoint for this analysis was the incidence of an elevated hs-TnT above the ULN if the baseline value was normal, or 1.5 ≥ times baseline if the baseline value was above the ULN within the first three months of treatment. The maximum measured value was taken into account [50].

The secondary key endpoints that were evaluated at baseline and at three months were as follows [50]:The incidence of hs-TnI and NT-proBNP above the ULN;Evolution of TTE parameters;Association between the evolution of troponin/NT-proBNP and TTE and electrocardiography parameters;Agreement between hs-TnT and hs-TnI levels.

### 4.5. Statistical Analysis

Study data were collected and managed using REDCap electronic data capture tools hosted at AZ Maria Middelares [52,53]. Statistical analysis was performed using IBM SPSS statistics 28.0 software (IBM Corporation, Armonk, NY, USA) and R Software version 4.1.3. Frequencies and percentages were reported for categorical variables. Continuous variables were described as mean ± standard deviation for those with a normal distribution. For non-normal distributed parameters, the median and interquartile ranges were noted. Where values were missing, percentages were calculated for the available cases, and the denominator was mentioned. The primary endpoint of hs-TnT elevation was studied in a competing risk framework, treating all-cause mortality as a competing event. Cumulative incidence and 95% confidence intervals were calculated. Cohen’s kappa (κ) was used to assess the agreement between hs-TnI and hs-TnT elevations, taking the ULN of each test into account. Only samples taken at baseline and at three months were used for this analysis. TTE parameters and NT-proBNP were compared at baseline and at three months using either a paired sample t-test, for normally distributed variables, or a Wilcoxon Signed Rank test for non-normally distributed variables. The level of statistical significance was set at *p* < 0.05.

## 5. Conclusions

It remains crucial to provide early evidence-based data on the role of cardiac biomarkers and TTE in the systematic follow-up of patients treated with ICIs to the cardio-oncological community, as the recent guidelines are still mainly based on expert opinions and clinical trials that have strict inclusion criteria, which does not reflect the real world cancer population. The early detection of subclinical CV dysfunction is needed to minimize the risks, reduce healthcare costs and keep patients on their life-prolonging therapy. Especially since ICIs are increasingly being administered in an early stage disease setting, where patients often have a better prognosis, side effects can significantly impact the patient’s quality of life. However, at present, there is no need for a more stringent follow-up than the current guidelines. Baseline measurements, on the other hand, should be performed in order to have an adequate reference value for each patient. Further enrollment in our study and a future pre-specified analysis will continue to elucidate the role of cardiac biomarkers and TTE, in both a larger group of participants and over an extended period of time.

In conclusion, this study provides new insights on the incidence of troponin elevations in ICI-treated patients and explores TTE values that could identify submyocardial damage in a uniform, Caucasian cohort. In addition, it is the first study to compare hs-TnT and hs-TnI in ICI-treated patients and to evaluate their interchangeability in the context of screening. Our preliminary analysis found hs-TnT elevations in 10.6% of cancer patients during the first three months of therapy in the absence of CV events. No significant changes were noted on 3D echocardiography nor in NT-proBNP at three months. The study is still ongoing, but these preliminary findings do not show a promising role for cardiac troponins nor for echocardiographic dynamics in the prediction of CV events during the early stages of ICI treatment.

## Figures and Tables

**Figure 1 pharmaceuticals-17-00965-f001:**
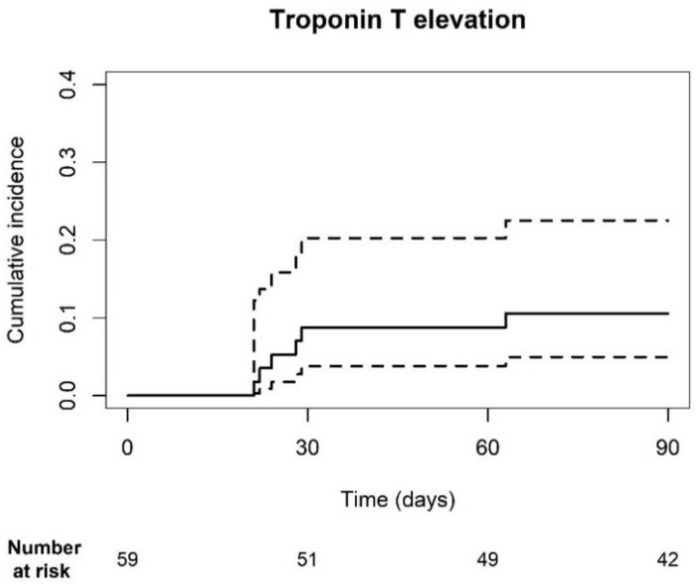
Cumulative incidence plot of troponin T elevation. The cumulative incidence of hs-TnT elevations was 10.6% [95% CI: 5.0; 22.5]. The dotted line represents the 95% CI.

**Table 1 pharmaceuticals-17-00965-t001:** Baseline characteristics.

	Parameters	Patient Cohort N = 59 (%)
General	ECOG 0 1 2 3	30 (51) 16 (27) 7 (12) 6 (10)
	Male gender	45 (76)
	Mean age (years)	68 ± 12
	BMI (kg/m^2^)	24.8 ± 4.1
Race/ethnicity	Caucasian	59 (100)
Primary tumor type	Renal cell carcinoma	15 (25)
	Bladder cancer	16 (27)
	Melanoma	14 (24)
	HNSCC	6 (10)
	NSCLC	2 (3)
	Merkel cell carcinoma	2 (3)
	Other	4 (7)
ICI regimen	Pembrolizumab	22 (37)
	Cemiplimab	1 (2)
	Avelumab	11 (19)
	Nivolumab	10 (17)
	Nivolumab + ipilimumab	15 (25)
Line of systemic treatment °	1st line	33 (56)
	2nd line	25 (42)
	≥3rd line	1 (2)
Risk factors	Diabetes mellitus	10 (17)
	Hypercholesterolemia	39 (66)
	Arterial hypertension	27 (46)
	Smoking status Current smoker Former smoker Never	8 (14) 29 (49) 22 (37)
	Prior chest irradiation	2 (3)
	eGFR < 60 mL/min	21 (36)
Cardiac history	Coronary artery disease *^-^	10/54 (19)
	Peripheral artery disease ^-^	9/52 (17)
	Stroke ^-^	5/58 (9)
	TIA	1 (2)
	Heart failure	3 (5)
	LVEF < 50% ^-^	5/54 (9)
	Calcium score > 400 ^+-^	14/52 (27)
Cardiac biomarkers	Hs-TnT (ng/L) Hs-TnI (ng/L) NT-proBNP (pg/mL)	11.6 [7.9; 19.5] 3.7 [2.5; 9.5] 148.0 [59.9; 413.0]
Medication	ACE-I/ARBs	17 (29)
	Beta blockers	22 (37)
	Diuretics	14 (24)
	Nitrate	1 (2)
	SGLT2-inhibitors	2 (3)
	Statins	29 (49)

ACE-I: angiotensin-converting enzyme inhibitor; ARB: angiotensin-2 receptor blocker; BMI: body mass index; eGFR: estimated glomerular filtration rate; ECOG: Eastern Cooperative Oncology Group; HNSCC: head and neck squamous cell carcinoma; hs-TnI: high-sensitivity troponin I; hs-TnT: high-sensitivity troponin T; ICI: immune checkpoint inhibitor; LVEF: left ventricular ejection fraction; NT-proBNP: N-terminal brain natriuretic peptide; SGLT-2 inhibitor: sodium–glucose cotransporter-2 inhibitor; TIA: transient ischemic attack. ° Patients receiving ICI treatment in the adjuvant setting were also noted under the first line of systemic treatment. Values are mean ± standard deviation or median [IQR_1_; IQR_3_]. * Coronary artery disease was defined as taking cardiac medication, prior coronary bypass surgery, or dilatation/stenting. ^+^ Calcium score was measured using a computed tomography scan in order to estimate the risk of heart disease based on calcium deposits in the coronary arteries. ^-^ For some parameters, data appeared incomplete or unknown. Therefore, we only described the proportion of patients for whom the data were available, along with the corresponding percentages.

**Table 2 pharmaceuticals-17-00965-t002:** Evolution of cardiac troponin I and NT-proBNP during the first three months of treatment.

Troponin I.	Baseline < 45.2 ng/L	Baseline ≥ 45.2 ng/L
3-month < 45.2 ng/L	48/49	0/49
3-month ≥ 45.2 ng/L	0/49	1/49
**NT-proBNP**	**Baseline < 125 pg/mL**	**Baseline ≥ 125 pg/mL**
3-month < 125 pg/mL	21/49	5/49
3-month ≥ 125 pg/mL	2/49	21/49

NT-proBNP: N-terminal prohormone brain natriuretic peptide.

**Table 3 pharmaceuticals-17-00965-t003:** Agreement in elevation between both cardiac troponins at baseline and three months.

	Baseline (*n* = 59)		3 Months (*n* = 49)	
Troponin	*Hs-TnI < 45.2*	*Hs-TnI ≥ 45.2*	*Hs-TnI < 45.2*	*Hs-TnI ≥ 45.2*
*Hs-TnT < 14.0 ng/L*	36	0	33	0
*Hs-TnT ≥ 14.0 ng/L*	20	3	15	1

hs-TnI: high-sensitivity troponin I; hs-TnT: high-sensitivity troponin T; *n* = number of patients.

**Table 4 pharmaceuticals-17-00965-t004:** Three-dimensional transthoracic echocardiography parameters at baseline and three months.

Parameters.	*n*	Baseline	3 Months	*p*
3D-LVEF (%)	44	56 ± 6	56 ± 6	0.90
LVEDV (mL)	43	105 [82; 129]	108 [94; 133]	0.25
LVESV (mL)	43	45 [34; 59]	48 [38; 63]	0.24
GLS (%) 2-chamber (%) 3-chamber (%) 4-chamber (%)	37 34 33 34	−17.8 [−18.5; −14.2] −16.92 ± 3.64 −18.0 [−19.0; −14.9] −16.81 ± 3.27	−17.0 [−18.8; −15.1] −16.84 ± 3.36 −17.0 [−18.8; −16.0] −16.59 ± 3.44	0.66 0.91 0.90 0.70
Right ventricular function TAPSE (mm) s’-wave (cm/s)	48 47	23 ± 5 12 ± 3	22 ± 4 13 ± 3	0.34 0.58
Dimensions Left atrial area (cm^2^)	47	17 ± 4	18 ± 5	0.26
Diastolic function E (cm/s) A (cm/s) E/A ratio Deceleration time (ms) Peak e’ velocity of septal wall (cm/s) Peak e’ velocity of lateral wall (cm/s) E/e’ septal wall E/e’ lateral wall MV E/e’ average	48 47 47 46 49 49 48 48 48	62 ± 16 76 ± 20 0.8 [0.7; 0.9] 194 ± 55 8 ± 2 9 ± 3 8 [6; 10] 7 [5; 8] 7 [6; 9]	63 ± 20 76 ± 17 0.8 [0.6; 1.0] 194 ± 62 8 ± 2 9 ± 3 8 [7; 10] 7 [5; 8] 7 [6; 9]	0.54 0.94 0.82 0.99 0.96 0.85 0.67 0.86 0.94

GLS: global longitudinal strain; LVEDV: left ventricular end-diastolic volume; LVEF: left ventricular ejection fraction; LVESV: left ventricular end-systolic volume; MV: mitral valve; TAPSE: tricuspid annular plane systolic excursion. *n* = number of patients. Values are mean ± standard deviation, or median [IQR_1_; IQR_3_].

## Data Availability

The datasets generated during and/or analyzed during the current study are available from the corresponding author on reasonable request.

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
