# Peer review of "Close Cardiovascular Monitoring during the Early Stages of Treatment for Patients Receiving Immune Checkpoint Inhibitors"

_pharmaceuticals, 2024, doi:10.3390/ph17070965_

Round 1

Reviewer 1 Report

Comments and Suggestions for Authors

The manuscript by Delombaerde and co-authors presents the results of clinical investigation where different biomarkers, including troponins I and T (hs-TnT and hs-TnI), for predicting cardiac adverse events were monitored in cancer patients receiving immune checkpoint inhibitors (ICIs). The main finding of the manuscript consists in the absence of cardiovascular events in patients, in spite of the hs-TnT elevation by 10.6%, and in a low correlation between hs-TnT and hs-TnI levels. These preliminary results are important and will be useful for further formulation of ICI-based therapeutic strategies. The manuscript deserves publication.

Specific comments:

Table 1:  The last column of the table indicates "Patient cohort N = 59 (%)".  However, some cells in this column have different format (Cardiac history, Cardiac biomarkers). Hence, additional explanations are needed in footnotes or in the column title.

Lines 187-190:  The sentences present in these lines are repeated in lines 193-196.

In my opinion, the manuscript can be accepted for publication after minor revision.

Reviewer 2 Report

Comments and Suggestions for Authors

In the study, authors aimed to provide insights on the incidence of troponin elevations and echocardiographic dynamics during ICI treatment in cancer patients and their role as a potential biomarker. The study is important in its topic however novelty of the manuscript should be explained clearly. It was mentioned that this manuscript is the first study to compare hs-TnT and hs-TnI in ICI treated patients and evaluate their interchangeability in the context of screening but mentioned these two biomarkers were compared in similar articles. (Petricciuolo, S., Delle Donne, M.G., Aimo, A., Chella, A. and De Caterina, R., 2021. Pretreatment highsensitivity troponin T for the shortterm prediction of cardiac outcomes in patients on immune checkpoint inhibitors. European Journal of Clinical Investigation51(4), p.e13400., Waissengein, B., Abu Ata, B., Merimsky, O., Shamai, S., Wolf, I., Arnold, J.H., Bar-On, T., Banai, S., Khoury, S. and Laufer-Perl, M., 2023. The predictive value of high sensitivity troponin measurements in patients treated with immune checkpoint inhibitors. Clinical Research in Cardiology112(3), pp.409-418. etc.) Also these articles could not find in reference section. The novelty and discussion section should be explained by adding similar articles.

Moreover the abstract of this article has already been published elsewhere.  (https://doi.org/10.1016/j.iotech.2023.100584). The information about mentioned abstract should be given in acknowledgement section.

Comments on the Quality of English Language

There are some typos which can be corrected.

Reviewer 3 Report

Comments and Suggestions for Authors

July 4, 2024

Ms. Ref. No.: pharmaceuticals-3096694

Journal: Pharmaceuticals.

Title: Close cardiovascular monitoring during the early stages of treatment for patients receiving immune checkpoint inhibitors.

Comments:

Thank you for your efforts in composing an on such a pertinent subject and many thanks for its valuable figures. I have taken the liberty of providing you with a few observations that I believe will serve to enhance the quality of your work. Please find my feedback outlined in the following paragraphs

1-      The first question is about, what is the incidence rate of cardiovascular disease due to immune checkpoint inhibitors?

2-      There are some additional biomarkers such as BNP, why was used troponin in this study?

3-      According to the result section of this article how can calculate the specificity of this biomarker (troponin)? And was this biomarker specified well for this disease?

4-      According to the result section of this article how can generalize the results?

5-      There are 59 patients in this study, how was determined this sample size?

6-      What was the main criteria for selecting patients here, or what was the inclusion and exclusion criteria?

7-      The sample size of men and women are not same, is it Ok?

8-      Is there any association between genus and age and BMI with results?

9-      Time frame of this study was three months, why this duration?

10-   If this study continue after three month, will be the results same or different?

11-   According to Table 1 there are some types of cancer between patients that their condition, risk factors and therapies maybe different to each other, can this factor influence the results? 

Round 2

Reviewer 2 Report

Comments and Suggestions for Authors

The manuscript can be accepted in its current form.

Reviewer 3 Report

Comments and Suggestions for Authors

No Thanks